# PTA-Welded Coatings with Saturation Magnetization above 1.3 T Using FeCrBSi Powders with Chemical Composition Similar to AISI 430 Ferrite Stainless Steel

Yingqing Fu [1,2,*], Haiming Wang [1], Wenhao Huang [3], Zhoujian Pan [1], Changhao Liu [1], Lei Zhao [1], Chao Li [1], Liangyu Zhu [1] and Naibao Huang [1,2]

1    Department of Materials Science and Engineering, Dalian Maritime University, Dalian 116026, China
2    Key Laboratory of Transport Industry of Ship-Machinery Maintenance and Manufacture,
     Dalian Maritime University, Dalian 116026, China
3    School of Mechanical Engineering, Dalian University of Technology, Dalian 116023, China
*    Correspondence: fuyingqing@dlmu.edu.cn

**Abstract:** Fe-Cr-based soft magnetic alloy (SMA) monolayer coatings with high saturation magnetization (Ms) above 1.3 T were deposited onto AISI 1010 substrate by co-axial powder feeding plasma transferred arc (PTA) welding, using FeCrBSi self-fluxing powders Fe313, which have a similar chemical composition to AISI 430 ferritic stainless steel (FSS). The effect of welding parameters on the phase assemblage, microstructure, hardness and magnetic performance of the coatings was investigated. The results show that the coating's maximum width and the welding surplus height increased with the rise in welding heat input and powder distribution density, respectively. The coating's Ms increased sharply, but its coercivity (Hc) decreased with the growth in the substrate dilution ratio. The coating's Hc increased whereas its Ms decreased with the increment in welding heat input. The as-welded coating C3 with optimum magnetic performance had a dendrites–eutectics composite structure, where the columnar or equiaxed sorbitic pearlite dendritic cores surrounded by network-like eutectics $\alpha$(Fe,Cr) + $(Fe_{1-x}Cr_x)_2B$ were the main contents. Moreover, $(Fe,Cr)_7C_3$ and CrB had also been detected, and they were mainly distributed in the interdendritic regions. The body-centered cubic (b.c.c.) $\alpha$(Fe,Cr) multi-element solid solution contributes to a high Ms of 1.61 T, and the borides $(Fe_{1-x}Cr_x)_2B$ and CrB as well as $(Fe,Cr)_7C_3$ and other carbides cause a high Hc of 58.6 Oe and hardness $HV_{0.3}$ of 4.90 ± 0.06 GPa, much higher than that of AISI 430 FSS (HV < 1.8 GPa). The current work verifies the feasibility of fabricating Ni- and Co-free FeCrBSi SMA coatings with high Ms and high hardness via PTA welding, and since the feedstock powders have chemical composition similar to AISI 430 FSS, the work may bring about novel applications for AISI 430 FSS in particular cases where the considerable wear-resistant performance as well as superior soft magnetic and anti-corrosive properties are required.

**Keywords:** soft magnetic alloy; saturation magnetization; coercivity; hardness; plasma transferred arc welding; coating; AISI 430

## 1. Introduction

Ferritic stainless steels (FSSs) without high-cost Ni as an alloying element have moderate corrosion resistance with lower material cost [1], and have soft magnetic properties with low coercivity (Hc), suitable saturation magnetization (Ms), high permeability, etc. An efficient method to minimize the eddy current losses in AC magnet applications is increasing electrical resistivity, and fortunately, FSSs have relatively higher electrical resistivity compared with soft iron and silicon steel. Furthermore, FSSs can be produced to large dimensions in some special applications, and their machinability is acceptable [2]. They also have a lower expansion coefficient compared with austenitic stainless steels, which is a significant advantage when temperature cycling resistance is required [3]. Considering

all of these features means that FSSs have been used in a wide range of magnetic applications [4]. Moreover, they can be applied in electrical motors, generators and relays [5]. Thus, AISI 430 stainless steel as a low-cost material for replacing austenitic stainless steel has received particular attention [1–3,6–8] due to its higher yield strength, higher ductility and better polarization resistance in harsh environments and its suitable soft magnetic properties.

However, soft magnetic alloys (SMAs) for electrical motors must be subject to severe mechanical loads and need high strength and ductility, and some magnetic devices need not only superior soft magnetic performance but also considerable wear resistance, such as magnetic poles in magnetic abrasive finishing [9], magnetic bearing [10] and magnetic fluid seal [11]. For these applications, traditional SMAs based on ferritic steels cannot meet the requirements alone. Multi-constituent SMAs with a ferromagnetic matrix [12,13] may address the issues mentioned above. Therefore, Ni- and Co-free self-fluxing [14] SMA Fe313 [15–17] powders with chemical composition similar to AISI 430 were used as feedstock in our current work, and Fe-Cr-based soft magnetic coatings with appropriately high Ms and much higher hardness than AISI 430, which are beneficial to lower material cost and higher wear resistance, are investigated in this paper.

Plasma transferred arc (PTA) welding is a rapidly growing method of surface cladding technology due to many advantages such as the low cost of equipment and operation and the free option of feedstock powders [18,19]. It has been confirmed to be a feasible and reliable way to process three-dimensional (3D) complex geometry parts based on experimental investigations [20], and can be used to produce SMAs [21,22], but the research on Fe-Cr-based SMAs fabricated by PTA welding is still scarce. Therefore, in the current study, the co-axial powder feeding PTA welding was adopted to prepare coatings using FeCrBSi self-fluxing powders with constituents similar to AISI 430 FSS, and the influence of welding parameters on the phase composition, microstructure, microhardness and magnetic properties of the coatings was investigated in detail. The current work may explore the feasibility of fabricating Co- and Ni-free Fe-Cr-based SMA coatings via PTA welding and propose new applications for AISI 430 FSS.

## 2. Experiment

### 2.1. PTA Welding Materials and Process

Quality carbon steel AISI 1010 was selected as a substrate with dimensions of $200 \times 80 \times 10$ mm$^3$. The feedstock was Fe-based self-fluxing powder Fe313 [15–17], whose X-ray diffraction (XRD) pattern, scanning electron microscope (SEM) images and energy dispersive X-ray (EDX) analysis results are presented in Figure 1. The powder phase assemblage contains a predominant phase, a multi-element b.c.c. $\alpha$(Fe,Cr) solid solution, and minor (Fe,Cr)$_7$C$_3$ (Figure 1a). Its constituent elements are Fe, Cr, B, Si, C, Mn and Ni (Figure 1a) and its chemical composition is similar to AISI 430 [23] FSS. The gas-atomized powders with a near-spherical shape (shown in Figure 1b,c) have a size of 120–325 screen mesh. Their chemical compositions are shown in Table 1. The upper surface of the substrate was prepared by surface grinding and cleaned thoroughly with absolute ethanol before deposition. The feedstock powders were dried in a vacuum oven, and then deposited onto AISI 1010 steel substrate with a PTA welding machine LU-F400-F300 [22]. High pure Ar gas was used as plasma, carrier and shielding gas, and the single-track PTA-welded coating was obtained with the parameters shown in Tables 2 and 3. The selection of the welding parameters was based on past experience and the equipartition method.

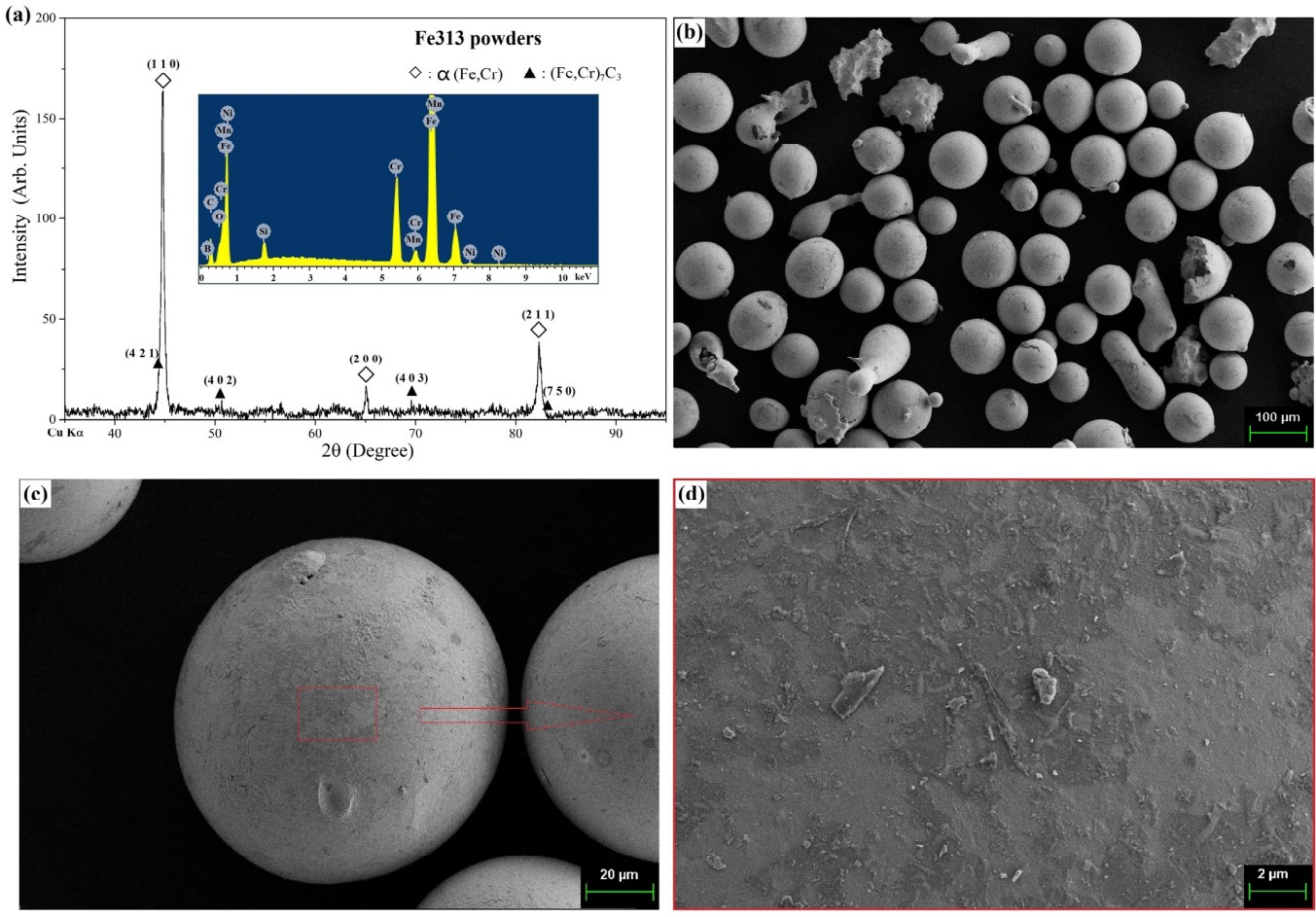

**Figure 1.** (**a**) XRD pattern. Inset: EDX spectrum showing peaks of each element. (**b**–**d**) SEM images for Fe313 feedstock powders presenting their overview and representative morphology.

**Table 1.** Chemical composition (wt.%) of substrate, Fe313 self-fluxing powders and AISI 430 FSS.

| Materials | C | Si | B | Cr | Ni | Cu | Mn | S | P | Fe |
|---|---|---|---|---|---|---|---|---|---|---|
| AISI 1010 | 0.07–0.14 | 0.17–0.37 | - | ≤0.15 | ≤0.25 | ≤0.25 | 0.35–0.65 | ≤0.04 | ≤0.35 | Bal. |
| Fe313 | 0.1–0.2 | 1.0–1.5 | 1.0–2.0 | 14.0–18.0 | 0.2–0.5 | - | 0.2–0.5 | - | - | Bal. |
| AISI 430 [23] | 0.12 | 1.00 | - | 16.0–18.0 | - | - | 1.00 | 0.03 | 0.04 | Bal. |

**Table 2.** Constant parameters during PTA welding process.

| Parameter | Value |
|---|---|
| Plasma gas (Ar) flow rate/(L·min$^{-1}$) | 7.5 |
| Carrier gas (Ar) flow rate/(L·min$^{-1}$) | 7.0 |
| Shielding gas (Ar) flow rate/(L·min$^{-1}$) | 8.0 |
| Stand-off distance/mm | 13 |
| Preheat temperature/°C | 25 |
| Internal diameter of coaxial nozzle D/mm | 6 |

**Table 3.** Varying parameters during PTA welding, welding surplus height H, coating maximum width B, volumetric dilution ratio from substrate Dv, cross-sectional hardness $HV_{0.3}$ and magnetic properties of the as-welded coatings.

| Sample No. | C-I | C-II | C-III | C1 | C2 | C3 |
|---|---|---|---|---|---|---|
| Welding current I/A | 75 | 85 | 95 | 100 | 100 | 100 |
| Transferred arc voltage U/V (average) | 22.5 | 22.5 | 22.5 | 25.1 | 26.3 | 26.8 |
| Powder feed rate F/(g·min$^{-1}$) | 16 | 16 | 16 | 20 | 16 | 12 |
| Welding speed S/(m·min$^{-1}$) | 0.16 | 0.16 | 0.16 | 0.14 | 0.18 | 0.22 |
| Powder distribution density $P_d$/(mg·mm$^{-2}$) | 16.7 | 16.7 | 16.7 | 23.8 | 14.8 | 9.1 |
| Welding heat input HI/(kJ·cm$^{-1}$) [$\eta = 0.7$] (average) | 4.4 | 5.0 | 5.6 | 7.5 | 6.1 | 5.1 |
| Welding surplus height H/mm | 1.7 | 1.5 | 1.2 | 4.3 | 2.2 | 1.5 |
| Maximum width B/mm | 5.3 | 4.9 | 4.8 | 12.1 | 9.5 | 8.2 |
| Volumetric dilution ratio $D_v$/% (average) | 0.6 | 3.6 | 7.6 | 4.97 | 18.78 | 32.50 |
| Microhardness $HV_{0.3}$/GPa (average) | 5.67 ± 0.17 | 5.83 ± 0.19 | 5.53 ± 0.19 | 4.87 ± 0.11 | 5.18 ± 0.09 | 4.90 ± 0.06 |
| Saturation magnetization Ms/T | 0.23 | 0.82 | 1.44 | 1.42 | 1.47 | 1.61 |
| Coercivity Hc/Oe | 113.4 | 70.0 | 109.8 | 76.4 | 74.2 | 58.6 |

## 2.2. Microstructure and Properties' Characterization

The as-welded coating samples were wire-electrical-discharge-machined and ground to have a uniformly flat surface or cross-section, then polished and etched. The etched specimens and feedstock powders were characterized by an Olympus GX51F computerized optical microscope (OM) and SEM (SUPRA55 SAPPHIRE, CARL ZEISS) with an EDX spectroscope (X-Max, OXFORD). XRD analysis was performed on the powders and coating surface XOY (plasma torch scanning plane) using an X-ray diffractometer (D/MAX-Ultima+/PC, Rigaku, Tokyo, Japan) with Cu Kα radiation ($\lambda = 0.15406$ nm) at a scanning speed of 8°/min. The coating microhardness $HV_{0.3}$ at a load of 2.94 N and a dwell time of 10 s was measured by an MH-6 hardness tester, and Vickers indentation marks were performed on 9 different locations on the coating cross-section YOZ plane (vertical to welding speed). A vibrating sample magnetometer (VSM) (7400-S, LakeShore, Westerville, OH, USA) under a maximum applied field of 15,000 Oe and a DC B-H loop tracer (MATS-2010SD, Linkjoin, Loudi, Hunan, China) were used to measure the values of Ms and Hc for the welded coating at room temperature. The measured magnetic field direction was perpendicular to the coating surface (i.e., direction Z).

## 3. Discussion

### 3.1. Microstructure

As shown in Table 3, six monolayer coatings were prepared by PTA welding, and the typical geometry characteristics of the coating cross-section are illustrated in Figure 2, including the coating maximum width (B), welding surplus height (H), melting depth (h), upper melting area ($A_P$) and bottom melting area ($A_S$).

In order to understand the effect of process parameters on coating geometry, the powder distribution density ($P_d$), welding heat input (HI) and substrate volumetric dilution ratio in the coating ($D_v$) were calculated (or roughly estimated) by Equation (1) [17], Equation (2) [24] and Equation (3) [24], respectively.

$$P_d = F/(S \cdot D), \tag{1}$$

where $P_d$, F, S and D represent powder distribution density, powder feed rate, welding speed and internal diameter of the coaxial nozzle, as shown in Tables 2 and 3.

$$HI = \eta \cdot U \cdot I/S, \tag{2}$$

where HI, I, S, U and η (shown in Table 3) represent welding heat input, welding current, welding speed, transferred arc voltage and thermal efficiency of the co-axial powder feeding PTA welding process (η = 0.7) [24], respectively.

$$D_v = A_S / (A_P + A_S), \tag{3}$$

where $D_v$, $A_S$ and $A_P$ represent volumetric dilution ratio (shown in Table 3), bottom melting area and upper melting area, as illustrated in Figure 2. Factually, the dilution ratio varies across coating thickness, and it is larger close to the interface adjacent to the substrate and declines towards the upper surface, so each coating's $D_v$ only denotes the average of the substrate dilution ratio. The PTA welding process conventionally has dilution ratios of 5–30% [25], and the $D_v$ values for most of the coatings (shown in Table 3) were approximately within the scope, except C-I and C-II. These two coating samples have inferior soft magnetic properties with relatively lower Ms (<1 T), due to their unsuitable process parameters. The welding current I and heat input HI for them are too low to produce quality coatings, as there is not enough energy to heat and melt feedstock powders and substrate adequately. As for the sample C-III, its Hc value is extremely high (>100 Oe) and could not meet the project requirements, though it has a high Ms of 1.44 T. Thus, these three coatings (C-I, C-II and C-III, shown in Table 3) are excluded from the following discussion.

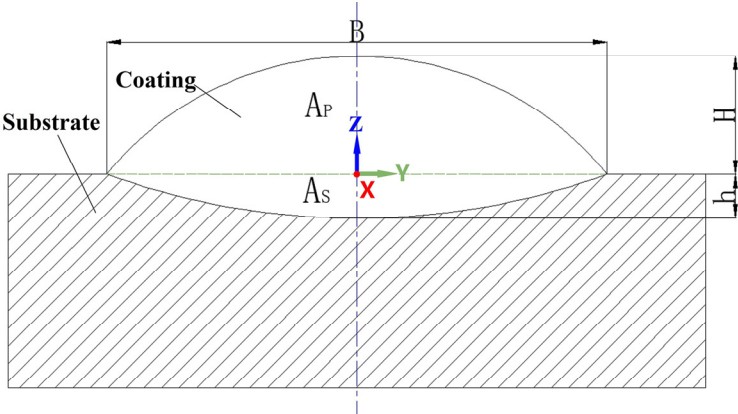

**Figure 2.** Typical geometry characteristics of the monolayer coating cross-section (B, H, h, $A_P$, $A_S$).

The effect of welding heat input (HI) and powder distribution density ($P_d$) on coating maximum width (B) can be seen in Figure 3. The coating maximum width increases with the increasing HI and $P_d$. According to the definition of welding heat input HI, it refers to the effective energy in the unit length along the welding direction (i.e., welding speed S), which is transferred to the powders and substrate, so it is also called welding linear energy. Therefore, it is not difficult to understand that with larger HI, more powders can melt in the unit length along S, and the extra powders which have not been melted are blown away, so the melting width, i.e., the coating maximum width B, is greater. Similarly, when HI is within the range where powders can be melted, the larger the powder distribution density $P_d$ is, the greater the amount of powders melted per unit area is, and the greater the value of B is. Obviously, the influences of both on B are similar.

The influences of HI and $P_d$ on welding surplus height (H) are shown in Figure 4. The welding surplus height H increases with the growth in HI and $P_d$. Hence, for the same reason mentioned above, with higher HI, more powders can melt per unit length along S, and the extra unmelted powders are blown away, so the welding surplus height H is greater. In a similar way, when HI is within the range where powders can be melted, the larger the powder distribution density $P_d$, the greater the amount of melted powders per unit area, and the greater the value of H. The influences of HI and $P_d$ on H are in the same manner.

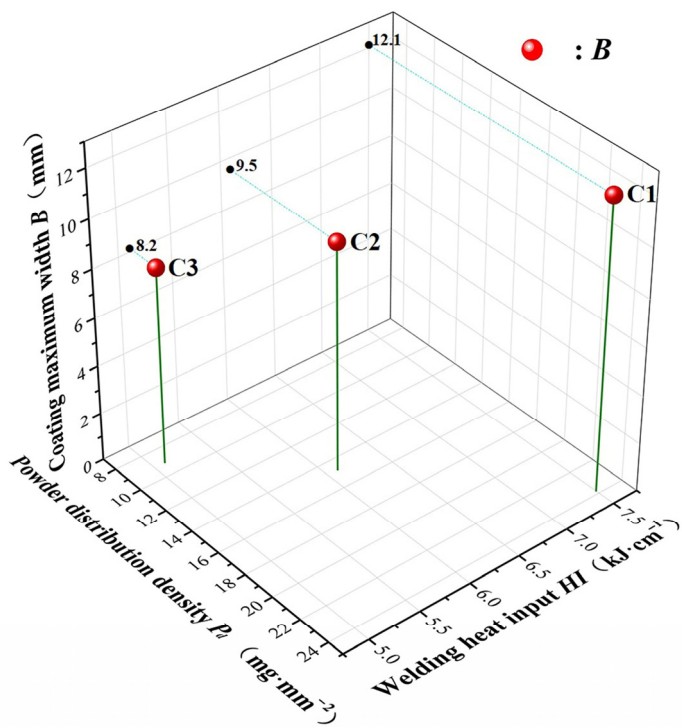

**Figure 3.** Influences of welding heat input (HI) and powder distribution density ($P_d$) on coating maximum width (B).

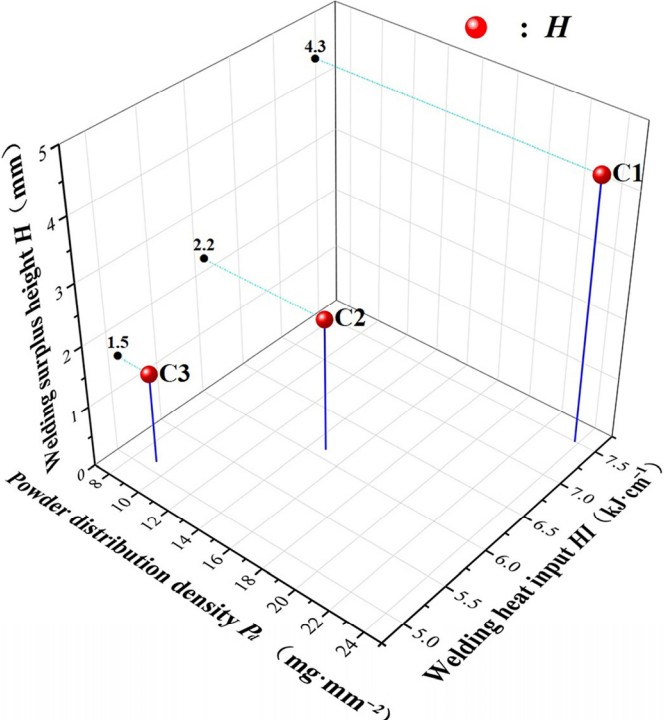

**Figure 4.** Influences of welding heat input (HI) and powder distribution density ($P_d$) on welding surplus height (H).

The overall cross-section morphologies of the coatings at a low magnification are shown in Figure 5a. The coating exhibits a uniform appearance without cracks and metallurgical bonding is formed between the coating and substrate. The coating's sharp XRD peaks shown in Figure 5b correspond to a high crystallinity, and are indexed as a representative b.c.c. phase that is verified to be ferritic α(Fe,Cr) solid solution. Moreover, it is

confirmed from the EDX results (Figure 1a) combined with XRD analysis that Cr, Mn, Ni, Si, B and C solute elements dissolved in this multi-element $\alpha$(Fe,Cr) solid solution. Not surprisingly, $(Fe,Cr)_7C_3$ which exists in the Fe313 powders is distinguished in the coatings. $(Fe_{1-x}Cr_x)_2B$, which was indexed as "$Fe_2B$" in Ref. [26] with a $CuAl_2$-type tetragonal structure, is not detected in the feedstock powders (Figure 1a) but identified in the coatings (Figure 5b). Compared with gas atomization of the powders, PTA welding of the coatings has a relatively slower cooling rate, which facilitates boron rejecting from $\alpha$(Fe,Cr) solid solution and the formation of $(Fe_{1-x}Cr_x)_2B$. For a similar reason, CrB precipitates undetected in feedstock powders are found in the coatings, which is consistent with Ref. [27].

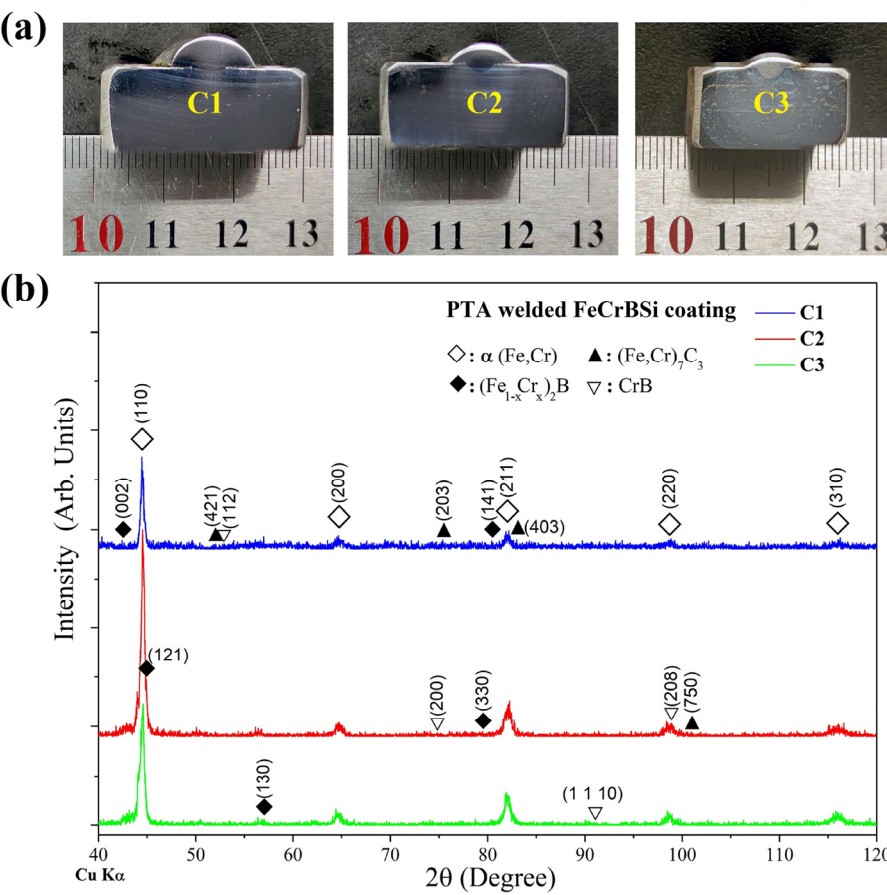

**Figure 5.** (**a**) Overall cross-section photographs at a low magnification and (**b**) XRD patterns of the PTA-welded coatings.

It can be observed from Figure 6 that the coatings have a typical rapid solidification structure with predominant dendrites, and the coating morphology varies across thickness. Each coating has three layers: the upper layer (shown in Figure 6a–c), middle layer (Figure 6d–f) and bottom layer or fusion layer [21] (Figure 6g–l). In the fusion layer, there is an interface (marked by the green word "Interface" between the green dashed line and the fusion line in Figure 6g–i) of about 200–300 μm thick, adjacent to the fusion line marked by white arrows in Figure 6g–l. The fusion line consists of planar grains, above which columnar cellular and even cellular dendritic grains for C1 (Figure 6g,j) and C3 (Figure 6i,l) or equiaxed grains for C2 (Figure 6h,k) exist. In the middle layer over this interface, the preformed columnar dendrites overlap and interlace with one another to form a dendrite network (Figure 6d–i), in which there are also minor equiaxed dendrites. With increasing distance from the fusion line, the columnar dendrite content declines gradually, and the equiaxed grain content increases [28,29]. Finally, the equiaxed dendrites almost totally dominate the upper layer of each coating (Figure 6a–c).

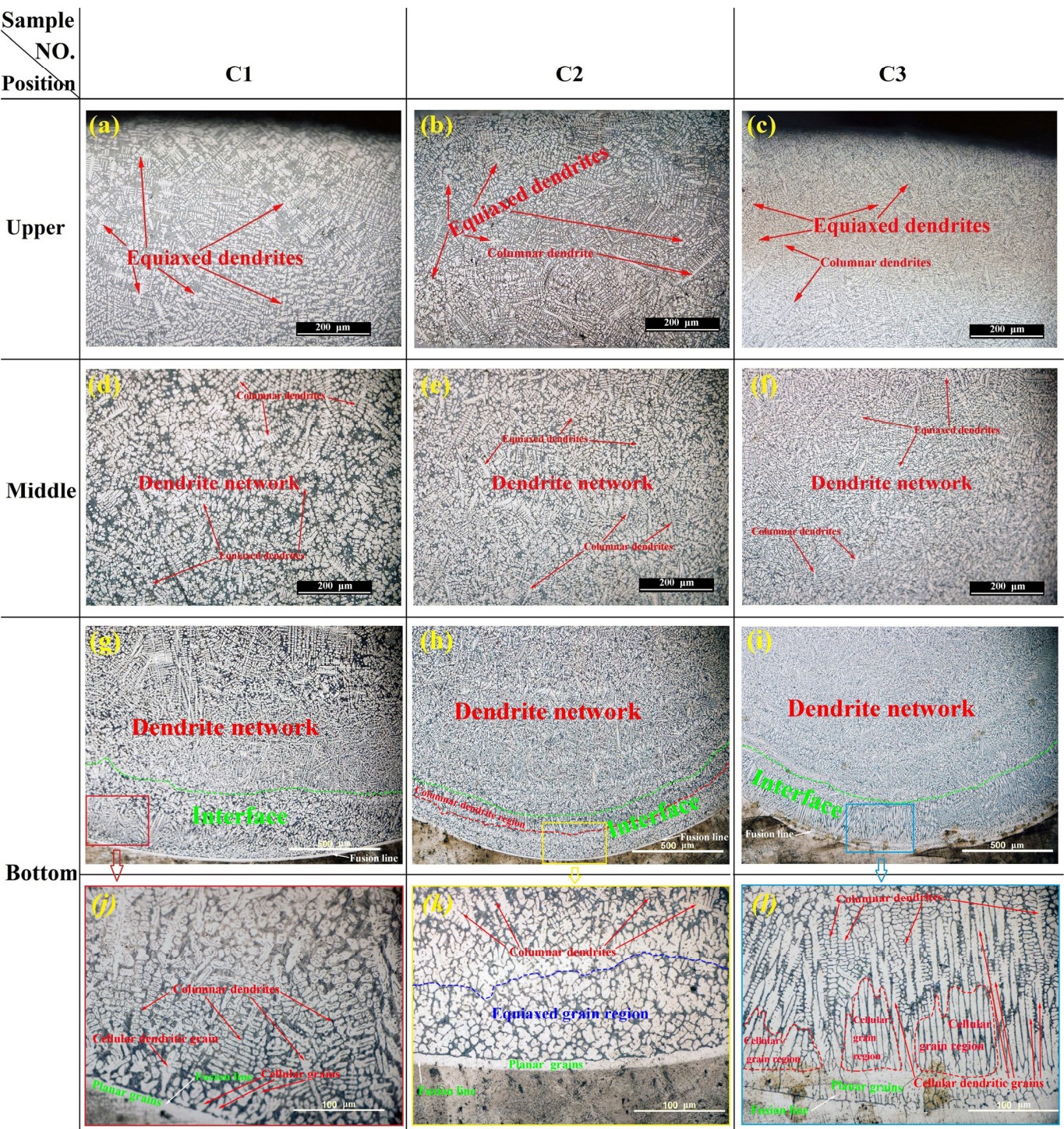

**Figure 6.** Optical micrographs of different zones in PTA-welded coatings (C1, C2 and C3): (**a**–**c**) upper microstructures of the coatings; (**d**–**f**) middle microstructures of the coatings; (**g**–**l**) bottom microstructures of the coatings.

The welded coating crystallization is mainly influenced by two factors: temperature gradient G and growth rate R [30]. The ratio of G to R dominates the solidification microstructure of the PTA-welded coating that can be planar, cellular or dendritic based on the solidification condition and the composition involved. At the beginning of solidification, a large temperature gradient G is produced in the interface between the liquid phase at the bottom of the welding molten pool and the cold substrate. The solid–liquid (S/L) interface

advances gradually towards the liquid phase and the G/R ratio increases, resulting in the formation of a layer of planar grains next to the substrate (Figure 6j–l), i.e., the fusion line. As the solidification proceeds, the G starts to diminish, while the R gradually increases, which leads to the increment in the constitutional supercooling (CS) in front of the S/L interface. Consequently, the stability of the planar grain boundary is destroyed, above which the typical columnar cellular and even cellular dendritic grains epitaxially grow for the coatings C1 (Figure 6j) and C3 (Figure 6l). However, non-dendritic equiaxed grains are formed above the fusion line in the coating C2, as shown in Figure 6k. Lippold et al. [31] have suggested that the non-dendritic equiaxed grain zone forms in a narrow temperature region adjacent to the fusion line by a heterogeneous nucleation mechanism aided by some high-temperature precipitates. M. Shakil et al. [32] have proved that the equiaxed cellular microstructure is highly dependent upon the alloy composition, turbulence and thermal gradients at the S/L interface and its formation is due to the uniform and symmetric cooling. As for the coating C2, compared with C1 and C3, its welding heat input HI of 6.1 kJ/cm is lower than that of C1 (7.5 kJ/cm) and higher than that of C3 (5.1 kJ/cm), i.e., $HI_{C1} > HI_{C2} > HI_{C3}$. The higher HI prolongs the high temperature retention time to slow down the cooling rate, which is helpful for grain growth. Thus, in the same position of the coating, the grain size (GS) of C1 is the largest and that of C3 is the smallest for these three coatings. In other words, C1 has the coarsest grains and C3 has the finest grains, as shown in Figure 6a–i. Moreover, the HI of C2 is between that of C1 and C3, and facilitates C2 to form a narrow temperature region, in which the non-dendritic equiaxed grains next to the fusion line can be formed, complying with a heterogeneous nucleation mechanism similar to Ref. [31], aided by $(Fe,Cr)_7C_3$, $(Fe_{1-x}Cr_x)_2B$ and CrB. With the G decreasing continually, the G/R ratio becomes smaller and the CS in front of the S/L interface becomes larger. Thus, the columnar dendrites begin to grow preferentially in a direction approximately vertical to the fusion line since the heat dissipation in this direction is the fastest. The mechanism of the dendrite structure is as follows: the formation of planar and cellular grains (including the non-dendritic equiaxed grains mentioned above) at the bottom of the coating causes the release of crystallization latent heat, which decreases the supercooling degree of the liquid phase, so the nucleation rate declines rapidly. Therefore, cellular grains tend to grow orientationally in a direction approximately perpendicular to the fusion line, in which the heat dissipation is the fastest. The grown grains form the long trunks of the columnar dendrites and secondary dendrite arms grow vertical to the trunks, as shown in Figure 6g–l. When the crystallization proceeds to a certain extent, the grown columnar dendrites overlap and interlace with one another to form dendrite networks (Figure 6d–i). The formation of columnar grains results from epitaxial grain growth along the direction normal to the substrate [33], i.e., the direction of the maximal thermal gradient in the welding molten pool [34]. In the subsequent crystallization stage, the top surface of the molten pool directly contacts the outside air, so that the G at the surface becomes smaller. The CS zone in front of the crystallization interface enlarges, which is beneficial to the nucleation and growth of new crystal nuclei. New surficial and endogenic heterogeneous nucleation sites surge and grow in random directions, restricting epitaxial growth for columnar grains and facilitating the formation of equiaxed grains [35]. Therefore, the fine equiaxed dendrites have become the predominant structures in the upper region near the coating surface (Figure 6a–c).

Figure 7 presents the high-magnification SEM micrograph and EDX elemental mapping of a typical dendrite microstructure of the coating C3, which consists of dendrites and interdendritic structures. Similar dendrite microstructures have also been reported in other research [36–38]. For instance, Lu et al. [37] reported that dendrites were firstly precipitated from the molten pool and then the intercrystalline metal was cooled to form netlike eutectics in the interdendritic regions. The EDX results (shown in Figures 7 and 8 and Table 4) reveal the composition segregation of the alloying elements in the coating. Fe, C and Si elements prefer to distribute in the dendrite more, while Cr and B elements tend to concentrate in the interdendritic area. The result that boron is more distributed in the interdendritic regions has also been observed by Gao et al. [39]. The composition segregation results from

the high cooling rate during welding, which restricts the solute atoms in the coating from sufficiently diffusing. Thus, the chemical compositions of the preformed dendrite and the interdendritic region are non-uniform. Additionally, the interaction between the alloying elements and surface tension of the molten pool, and the convection in the molten pool attributed to the uneven distribution of the plasma arc energy density, will also influence the composition segregation [40].

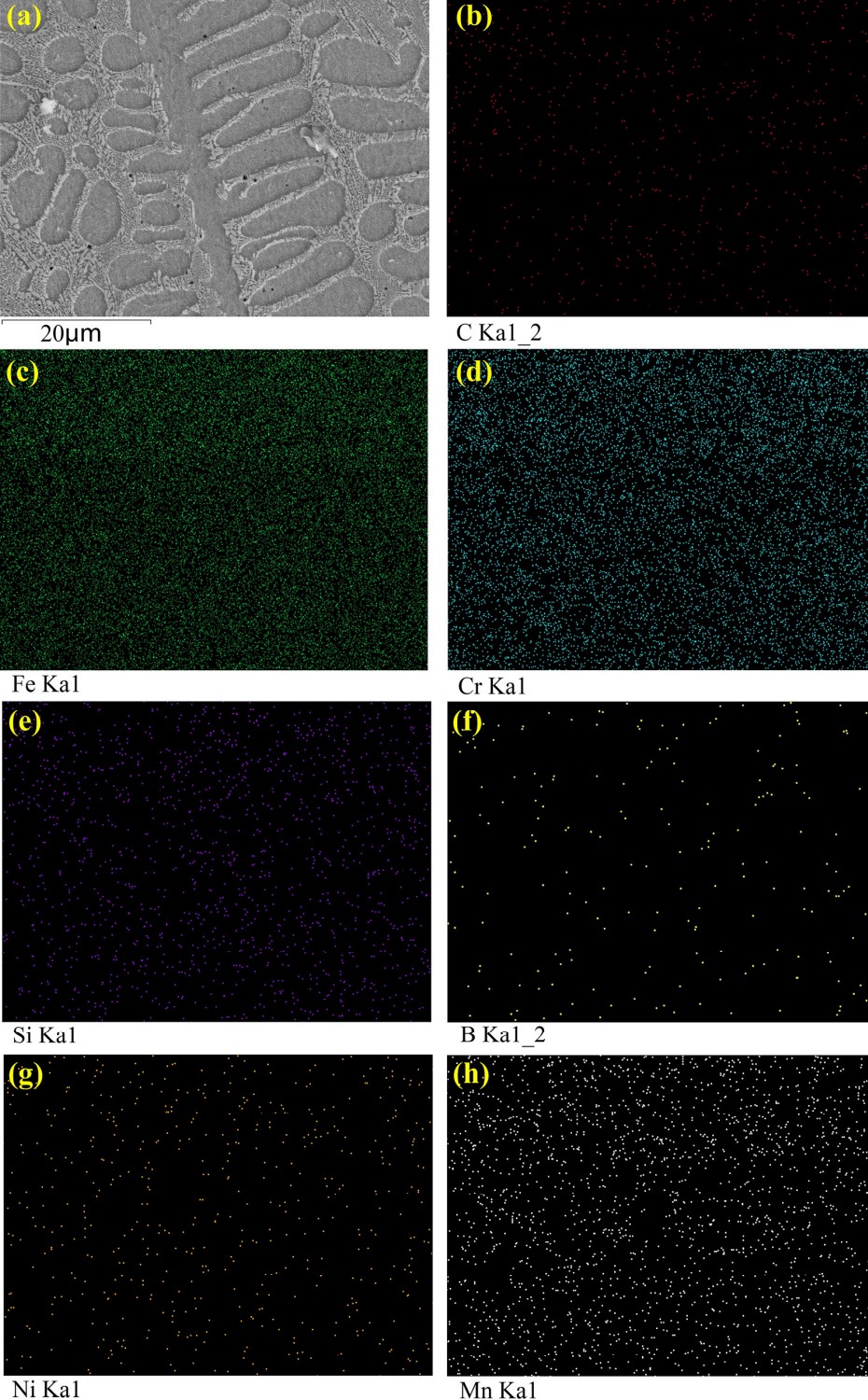

**Figure 7.** SEM micrograph (**a**) and the corresponding element distributions (**b**–**h**) of a typical dendrite structure for the coating C3.

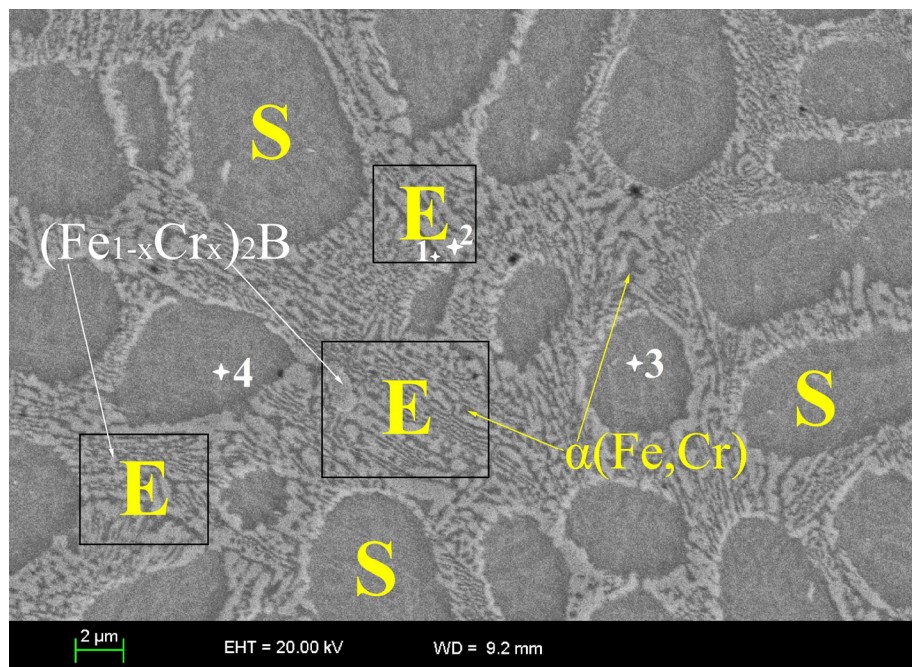

**Figure 8.** Higher-magnification SEM micrograph for the coating C3.

**Table 4.** EDX point scanning results of points (denoted by the white asterisks in Figure 8) in the coating C3.

| Elements | Weight Percent (wt.%) | | | |
|---|---|---|---|---|
| | Point 1 | Point 2 | Point 3 | Point 4 |
| Fe | 66.46 | 61.54 | 77.59 | 65.77 |
| Cr | 14.18 | 16.26 | 12.63 | 11.05 |
| B | 11.76 | 13.13 | - | - |
| Si | 0.65 | 0.51 | 1.32 | 1.34 |
| C | 6.95 | 8.07 | 8.07 | 21.47 |
| Mn | - | 0.49 | 0.39 | 0.36 |
| Ni | - | - | - | - |

It can be observed that the PTA-welded coating has a dendrites–eutectics composite structure, in which the columnar or equiaxed sorbite dendrite cores (denoted by "S") are surrounded by the network-like eutectics (marked by "E"), as shown in Figures 7 and 8. According to the Fe-B phase diagram [41], the austenite + $Fe_2B$ eutectic exists at 3.8 wt.% boron, while the average boron content in the PTA-welded coating conforms to a hypoeutectic constitution, containing primary austenite ($\gamma$-Fe) and an eutectic ($\gamma$-Fe + $Fe_2B$). Similarly, the coating corresponds to a hypoeutectoid steel composition, consisting of proeutectoid ferrite and pearlite. Combined with the phase diagrams [41] and XRD results, SEM and EDX analyses show that this ultrafine eutectic structure in the interdendritic region has a ferritic matrix of $\alpha$(Fe,Cr) solid solution with inclusions of $(Fe_{1-x}Cr_x)_2B$, denoted in Figure 8 by yellow and white arrows, respectively. The dendritic islands in the coating are sorbitic pearlites, indicated by "S" in Figure 8. They are transformed from primary austenite $\gamma$-Fe (rich in carbon) at room temperature [42], and the formation of these finer pearlites, called sorbitic pearlites [43,44], may be attributed to the relatively faster cooling rate of PTA welding (compared with conventional casting).

### 3.2. Hardness

The average microhardness (HV$_{0.3}$) values of the coating cross-sections are shown in Table 3, which are higher than those of AISI 1010 substrate [22] and AISI 430 FSS (HV < 1.8 GPa) [23], due to the high solid solubility of B, C, Si, Ni and Mn, as well as (Fe$_{1-x}$Cr$_x$)$_2$B, (Fe,Cr)$_7$C$_3$ and CrB hard phases included in the α(Fe,Cr) solid solution matrix, and uniform fine crystalline grains. It is well reasoned that the coating hardness declines with the increasing substrate dilution ratio D$_v$, since the rise in the dilution ratio of relatively soft AISI 1010 substrate with much lower hardness (<1.8 GPa) reduces the volume fraction of hard phases and the solute content in α(Fe,Cr) solid solution. As well as the substrate dilution ratio D$_v$, the welding HI has a considerable effect on the coating hardness. With the rise in HI, the energy for melting feedstock powders and substrate increases, and the cooling rate of molten pool becomes slower, promoting grain growth and solute (Ni, Mn, Si, B, C) precipitation from α(Fe,Cr) solid solution, and decreasing the coating hardness. However, increasing HI also facilitates the formation of hard phases such as (Fe$_{1-x}$Cr$_x$)$_2$B, (Fe,Cr)$_7$C$_3$ and CrB, leading to the coating hardness increment. The effect of the former surpasses that of the latter on the coating hardness, under the experimental conditions in this paper. Thus, the hardness of the coating with a lower substrate dilution ratio D$_v$ (<20%) decreased with the rise in HI, and the influence of D$_v$ on its hardness can nearly be omitted. In other words, the lower the HI, the higher the coating hardness, as shown in Table 3 (C-I, C-II, C-III, C1 and C2). As for C3, the influence of its high D$_v$ (32.5%) on its hardness cannot be neglected, and it prevailed over that of HI, so the hardness of C3 is not higher but lower than that of C2, though C3 has lower welding HI. Therefore, C2 has the highest hardness, whereas C1 and C3 have similar hardness values that are lower than that of C2 (Table 3 and Figure 9).

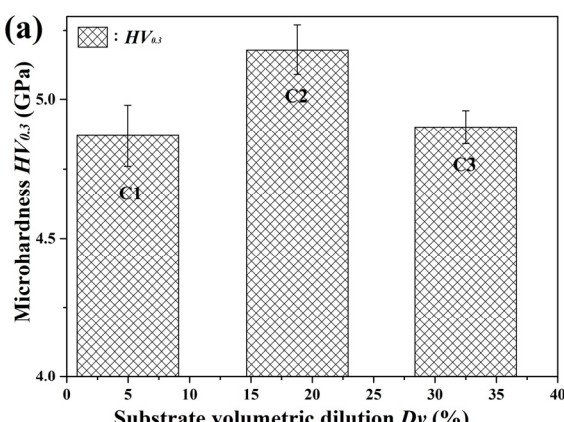 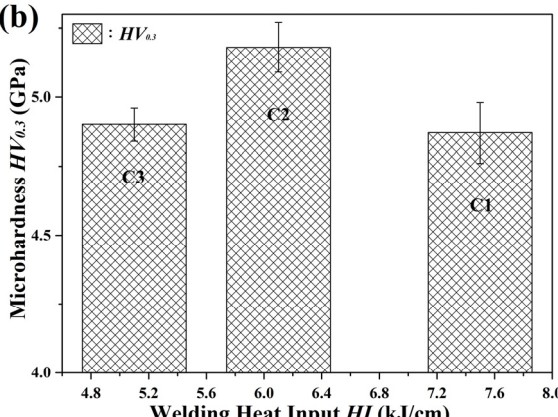

**Figure 9.** Correlation (**a**) between microhardness HV$_{0.3}$ and substrate volumetric dilution ratio D$_v$, and (**b**) between microhardness HV$_{0.3}$ and welding heat input HI for the PTA-welded coatings.

### 3.3. Magnetic Properties

The magnetic properties of the feedstock powders and the PTA-welded coatings, with Hc and Ms measured by a VSM, are shown in Table 3 and Figure 10. The coatings' VSM sample cuboids (2.0 mm × 2.0 mm × 1.5 mm) were cut from the welded overlayer by wire electrical discharge machining, as shown in Figure 10b, and they contained only the coating without the substrate.

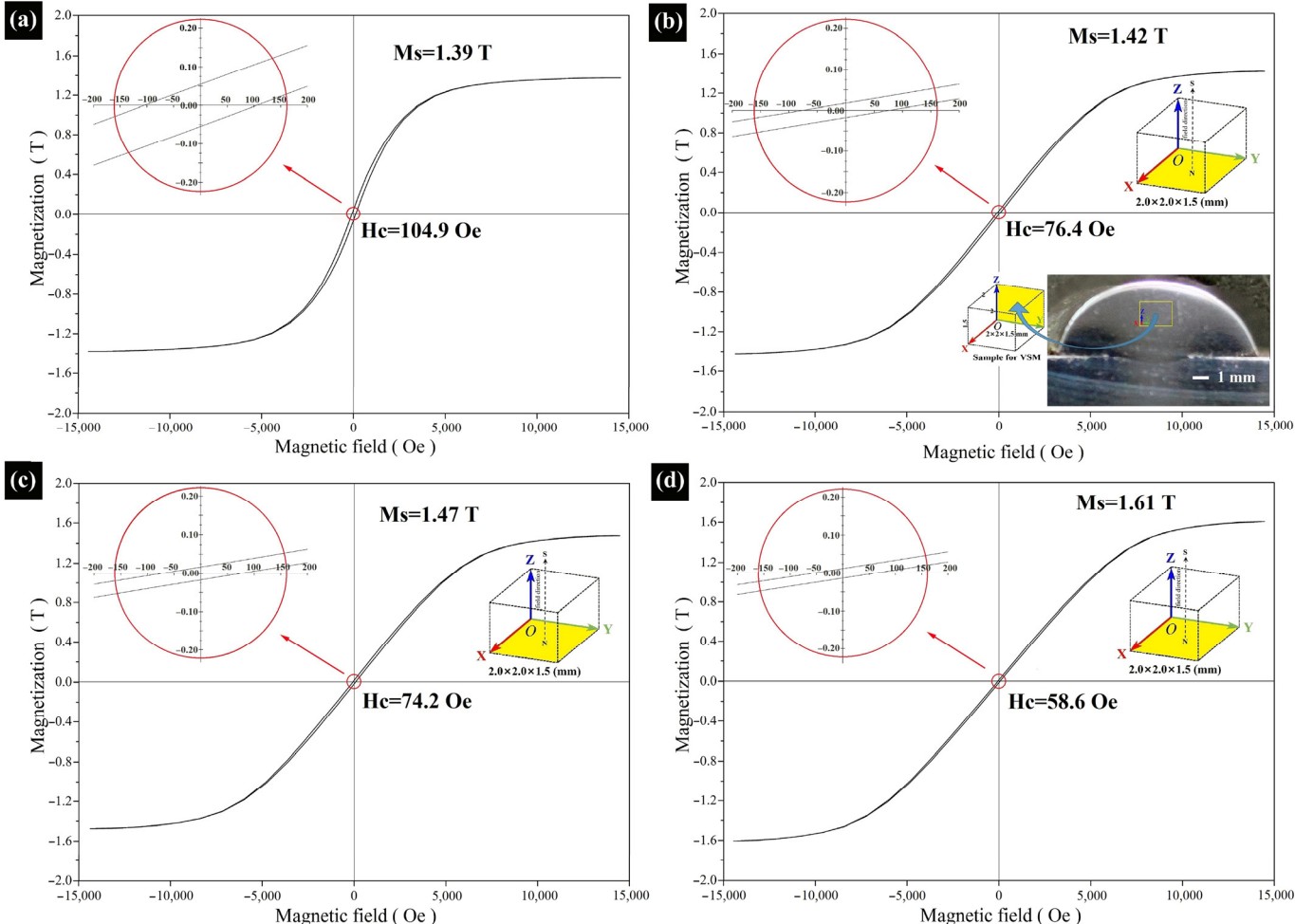

**Figure 10.** Hysteresis loops of (**a**) the feedstock powders Fe313 and the PTA-welded coating (**b**) C1, (**c**) C2 and (**d**) C3 sorted by increasing saturation magnetization Ms.

From the hysteresis loops (Figure 10), the powders and the PTA-welded coatings present adequate soft magnetic performance with high Ms above 1.3 T. Compared with the powders with Ms of 1.39 T and Hc of 104.9 Oe (Figure 10a), all the coatings have lower Hc and higher Ms (Figure 10b–d), i.e., the coatings have soft magnetic properties superior to the feedstock powders. Among all the coatings, C3 has the best soft magnetic properties with the lowest Hc of 58.6 Oe and the highest Ms of 1.61 T (Figure 10d).

The coercivity of soft magnetic materials depends on many structural features, such as phase composition, volume fractions of ordered and disordered phases, the critical size of magnetic single-domain state and the size of magnetically independent grains. Conventionally affected by most defects such as precipitates, dislocations and grain boundaries, the coercivity Hc relies on grain size, as in the following Equation (4) [45]:

$$H_c \approx 3\sqrt{\frac{k_B T_c K_1}{a M_s}}\frac{1}{D} \tag{4}$$

where Hc, $k_B$, Tc, $K_1$, a, Ms and D represent the coercivity, Boltzmann constant, Curie temperature, magnetocrystalline anisotropy, lattice constant, saturation magnetization and grain size, respectively.



The generally used correlation between coercivity and non-magnetic or low-Ms (compared with $\alpha$-Fe) inclusions is given in Equation (5) [45], following Kersten's theory.

$$H_c \propto \frac{\delta_\omega K_1}{M_s \mu_0 r} V_f^{2/3} \tag{5}$$

where $\delta_w$, $K_1$, Ms, $\mu_0$, r and $V_f$ represent the magnetic domain wall thickness, magnetocrystalline anisotropy, saturation magnetization, permeability of vacuum, average radius and volume fraction of the non-magnetic or low-Ms inclusions. Equation (5) does not take effect if $\delta_w \gg r$ or $\delta_w \ll r$. For the current welded coatings, the magnetic domain wall thickness $\delta_w$ can be estimated to be about 40–250 nm by referring to soft iron (~0.1 μm) [4] and silicon iron [46]. The particle size of most "low-Ms" (compared with $\alpha$-Fe) inclusions in the welded coatings was estimated to be about 20~2000 nm [26,44,47,48], i.e., r is comparable to $\delta_w$ (40–250 nm), so Equation (5) can be used to evaluate the influence of low-Ms inclusions (such as $(Fe_{1-x}Cr_x)_2B$, $(Fe,Cr)_7C_3$ and CrB) on the coating's Hc.

The solubility limit of Cr, B and even C in $\alpha$-Fe solid solution results in precipitating out low-Ms particles, accompanied by a decline in Ms and an increase in Hc. The relatively slower cooling rate of PTA welding releases internal stress induced by feedstock powder gas atomizing, and facilitates grain growth as well as the precipitation of low-Ms inclusions. The larger grains inhibit domain wall motion less effectively, leading to softer magnetic properties, whereas domain wall motion is resisted by these low-Ms inclusions, leading to harder magnetic properties [2]. The internal stress release and grain growth decrease Hc based on Equation (4), while low-Ms precipitates increase Hc according to Equation (5). The influence of the former on Hc surpassed that of the latter, and thus all the coatings have lower Hc than that of the feedstock powders (Figure 10).

Nevertheless, for the welded coatings (C1, C2, C3) prepared with different welding HI, the coating with larger grains unexpectedly has higher Hc, as shown in Table 3, Figures 6 and 11. This may be explained as follows. As mentioned above, the rise in PTA welding HI prolongs the high temperature retention time of the molten pool to slow down the cooling rate, which is helpful for grain growth and the precipitation of low-Ms inclusions. So, in the same position of the coating, C1 has the coarsest grains and C3 has the finest grains, as shown in Figure 6a–i. Meanwhile, C1 has more low-Ms precipitates than C2 and C3, and C3 contains the fewest low-Ms inclusions. For the current welded coatings, the effect of low-Ms precipitates on Hc prevails over that of grain growth. Moreover, the increase in the substrate dilution ratio $D_v$ can decrease the volume fraction of low-Ms inclusions $V_f$, leading to the decline in Hc according to Equation (5), as illustrated in Figure 11b. Hence, the higher the welding HI, the larger the grains and the higher the Hc for the coating, as shown in Table 3, Figures 6 and 11a.

As shown in Figure 10, it is revealed that the saturation magnetization Ms was increased after PTA welding, i.e., all the coatings have higher Ms than the feedstock powders. The formation of grains in the welded coating which are preferably oriented in the easiest direction of magnetization may be the main reason for the increase in Ms [2]. The maximum possible Ms of a ferromagnetic material represents the magnetization that results when all the magnetic dipoles in a solid piece are mutually aligned with the external field. The Ms of magnetic materials depends on the element composition, the number of magnetic atoms, the crystal structure and magnetic moment of atom [45]. It was mentioned above that there were some low-Ms precipitates distributed in a ferritic matrix in the PTA-welded coatings. These low-Ms inclusions decrease the coating Ms, and the reduction in volume fraction of these inclusions or the absence of these inclusions during PTA welding may increase the coating Ms, which is consistent with Ref. [2]. Thus, the coating Ms increases with the rise in the substrate volumetric dilution ratio $D_v$, resulting from the decrease in volume fraction of these low-Ms inclusions (Figure 11b). Furthermore, with increasing chromium content in the material, its Ms declines, which has previously been reported for ferritic stainless steels [49] and for iron chromium alloys [50], and in both cases, a linear decrease in Ms was observed. In other words, the Ms increases with the reduction in chromium

content in the material. The increase in the substrate volumetric dilution ratio $D_v$ also promotes decreasing chromium content in the coating, leading to the rise in Ms. Hence, the Ms of the coating increases sharply with the growth in $D_v$ (Figure 11b). On the other hand, with increasing welding HI for the coating, low-Ms precipitates proliferate, leading to the decrease in the Ms, shown in Figure 11a.

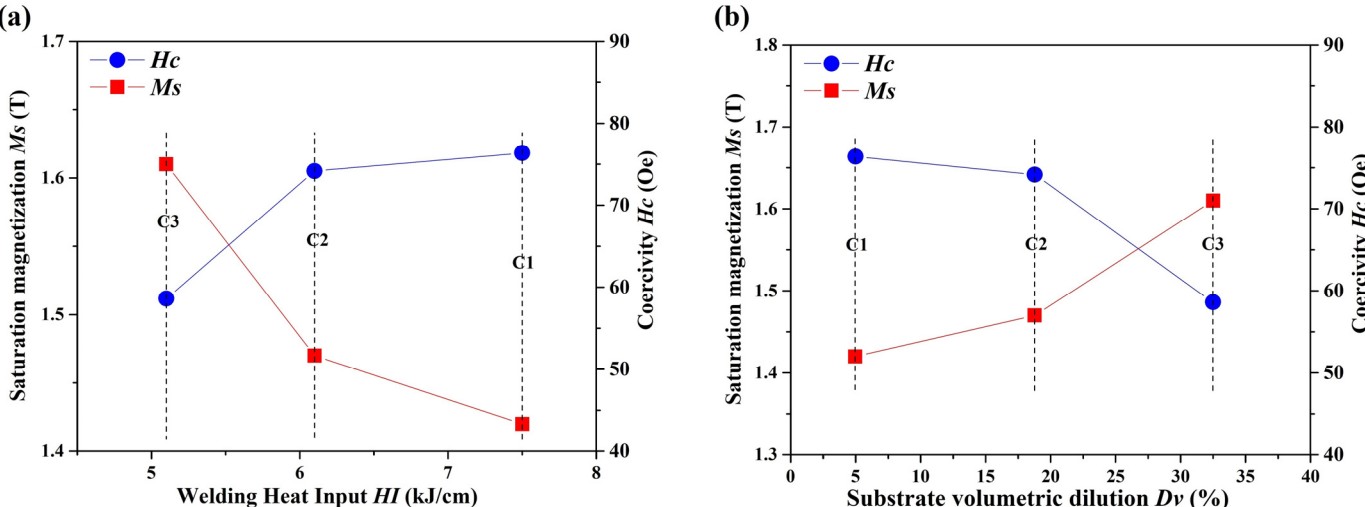

**Figure 11.** Correlation (**a**) between saturation magnetization Ms (or coercivity Hc) and heat input HI, and (**b**) between Ms (or Hc) and substrate volumetric dilution ratio $D_v$ for the PTA-welded coatings.

In summary, all the PTA-welded coatings (C1, C2 and C3) present adequate soft magnetic properties superior to the feedstock Fe313 powders, which are attributed to the existence of b.c.c. $\alpha$(Fe,Cr) solid solution with larger grain size. The Ms is from 1.42 T to 1.61 T and the Hc is from 76.4 Oe to 58.6 Oe. Among all the coatings, C3 had the highest Ms of 1.61 T and the lowest Hc of 58.6 Oe, and the optimal values for the powder flow rate, welding current and welding speed were 12 g/min, 100 A and 0.22 m/min, respectively. The precipitation of $(Fe_{1-x}Cr_x)_2B$, $(Fe,Cr)_7C_3$ and CrB seems to be detrimental for soft magnetic properties [45], but is helpful for the increase in hardness, so the coatings have hardness (HV > 4.50 GPa) much higher than that of AISI 430 FSS (HV < 1.8 GPa) [23], which may extend their applications in some special working conditions where appropriate wear-resistant and anti-corrosive properties as well as suitable soft magnetic performance are required.

## 4. Conclusions

1.  The high-crystallinity FeCrBSi monolayer coatings were fabricated by PTA welding using Fe313 self-fluxing powders with constituents similar to AISI 430 FSS. The welded coating has hypoeutectic structures composed of columnar or equiaxed dendrites of sorbitic pearlites and interdendritic network-like $\alpha$(Fe,Cr) + $(Fe_{1-x}Cr_x)_2B$ eutectics, with hardness (HV > 4.5 GPa) much higher than that of AISI 430 FSS (HRB < 88 or HV < 1.8 GPa) [23] and high Ms (>1.3 T), which are not lower (or even higher) than those of the ferritic steels with the same or similar nominal compositions (1.1–1.6 T) [4,49]. This may broaden the application for AISI 430 FSS in some special fields requiring not only fair corrosion and wear resistance but also superior soft magnetic properties.
2.  All the welded coatings (C1, C2 and C3) have superior soft magnetic performance than the original powders, i.e., the coatings have lower Hc and higher Ms than those of the Fe313 powders. The decrease in Hc is attributed to the formation of larger grains, and the grains in the PTA-welded coatings are preferably oriented in the easiest direction of magnetization, resulting in the increase in Ms.

3. Under the experimental conditions in this paper, the coating maximum width and the welding surplus height increase with the rise in welding HI and $P_d$, respectively. The coating's Hc decreases whereas its Ms increases sharply with the growth in $D_v$, and the former increases but the latter decreases with the increasing welding HI. The coating C3 presents the best soft magnetic properties with Ms of 1.61 T and Hc of 58.6 Oe. The coating's highest Ms is due to its highest $D_v$, lowest $V_f$ of non-magnetic or low-Ms (compared with $\alpha$-Fe) inclusions and lowest Cr content. The coating's highest $D_v$ and lowest $V_f$ also are responsible for its lowest Hc.

4. The high Hc of 58.6~76.4 Oe for the coatings, which resulted from the residual stress and deformation caused by the relatively rapid solidification of the PTA welding process, may be decreased by the further optimization of welding parameters or post-weld vacuum annealing, and these issues will be investigated in our follow-up research.

**Author Contributions:** Formal analysis, H.W., Z.P., C.L. (Changhao Liu), L.Z. (Lei Zhao), C.L. (Chao Li), L.Z. (Liangyu Zhu) and Y.F.; data curation, H.W., Z.P., C.L. (Changhao Liu) and L.Z. (Lei Zhao); H.W., Y.F. and N.H.; funding acquisition, investigation, supervision, project administration, Y.F. and N.H.; investigation, resources, W.H. All authors have read and agreed to the published version of the manuscript.

**Funding:** This research was funded by National Key Research and Development Program of China (2022YFB4002102) and National Defense Pre-Research Foundation of China (61409230612).

**Institutional Review Board Statement:** Not applicable.

**Informed Consent Statement:** Not applicable.

**Data Availability Statement:** The data that support the findings of this study are available from the corresponding author upon reasonable request.

**Acknowledgments:** The authors are grateful for the support of Dalian University of Technology for magnetic properties characterization.

**Conflicts of Interest:** The authors declare no conflict of interest.

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
