# Peer review of "PTA-Welded Coatings with Saturation Magnetization above 1.3 T Using FeCrBSi Powders with Chemical Composition Similar to AISI 430 Ferrite Stainless Steel"

_magnetochemistry, doi:10.3390/magnetochemistry9040093_

Round 1

Reviewer 1 Report

Dear Editor and Authors,

The paper "PTA-welded coatings ... similar to 3 AISI 430 ferrite stainless steel" is devoted to the deposition, structure, and static magnetic properties of a Fe-based protective coating. Plasma-assisted deposition technique based on welding is used. The paper is well-written, informative, and instructive. It fits well within the Aims and Scope of the MDPI Magnetochemistry Journal. This manuscript is definitely to be accepted for publication. However, I recommend considering three remarks, that can make this good paper even better. Thus, I recommend a minor revision.

1. The only thing I did not like in the manuscript is the first paragraph of the Introduction section. The intro should have been written clearly. I needed re-read this paragraph 3 times to get what you were after.

Starting from the first sentence:

-Ferritic stainless steels (FSSs) don’t contain any nickel as an alloying element have moderate corrosion resistance with lower...
-Ferritic stainless steels (FSSs) THAT don’t contain any nickel as an alloying element have moderate corrosion resistance with lower...?
-Ferritic stainless steels (FSSs) don’t contain any nickel as an alloying element THAT IS WHY THEY have moderate corrosion resistance with lower...?

The second sentence:
-An essential factor to minimize the eddy current losses in AC magnet applications
-Is it typical for FSS to be applied as AC magnets?

Further on:
-They also have lower expansion coefficient compared with austenitic stainless steels
-Why the austenitic stainless steel? Is it good in AC magnet applications?

Some more:
-Thereinto, AISI 430 stainless steel as a low
-Is AISI 430 in the FSS class?

And more:
-Therefore, Ni and Co-free self-fluxing [14] SMA Fe313 [15–17] powders with chemical composition similar to AISI 430 was used as feedstock in our current work, and Fe-Cr based soft magnetic coatings with appropriately high Ms and much higher hardness than that of AISI 430, which are beneficial to lower material cost and higher wear resistance, will be investigated in this paper.
-I did not get here, what you were after. Did you want to obtain some coating better than AISI 430 on top of the soft magnetic metal? Does the quality carbon steel AISI 1010 meet these requirements? Is it important for the substrate to be magnetic?

I think a clearer structure of the first paragraph is required here.

2. In the XRD study section it is not clear, what and how exactly you were studying. Did you divide the coating and the substrate? If not, why do you postulate that the XRD pattern is from the coating but not from the substrate?

3. The same remark is for the VSM section. You give precise meanings: Ms of 1.39±0.003T and Hc of 104.9±0.6Oe. It is not clear, how you obtained these low errors. I guess you just studied one sample and took the errors from one measurement. In my vast experience, VSM errors are rarely lower than 10% of the measured value... This is because there are several operations involved in the formation of this error: precise definition of the shape of the sample, weighting, and sample positioning... Related to this problem: what was the spatial uniformity of the properties of coating?

Reviewer 2 Report

This paper reports the influence of welding parameters on the structure and performance of PTA-welded FeCr-based soft magnetic alloy coatings. The findings are interesting. The comments below must be addressed in the revised manuscript before the paper can be considered for publication.

1. Abstract and Conclusions should be rewritten. The author spent a lot of space in the text to discuss the influence of preparation parameters,e.g., welding heat input and powder distribution density, but there is no mention of this part in the abstract and conclusion. “high saturation magnetization (Ms) above 1.3 T” in the abstract and the performance of the coatings is compared with AISI 430 FSS, which is mentioned in Introduction, however, in conclusions, it is said “high saturation magnetization Ms (> 1.4 T)” and compared with FeCo-2V soft magnetic alloy. What exactly is the purpose of the scientific work in this paper? The logic in the abstract should be reorganized and make the novelty of the work more clear. The same goes for the conclusion.

2. Please provide the basis for the selection of welding parameters.

3. If C-I, C-II and C-III are excluded from the discussion, why did the authors include them in the paper?

4. The figures need to be modified. Many details are not clear. The elements in EDX spectrum of Figure 1, the scale bar and colored words in Figure 6, numbers in the inserts of Figure 10.

Reviewer 3 Report

I recommend to publish the manuscript after minor revision. Y. Fu and coworkers report on the feasibility of fabricating FeCrBSi SMA coatings with high saturation magnetization and high hardness via PTA welding. Please see my comments and suggestions given below.

-          In line 31, I suggest to modify as: “do not contain any nickel as an alloying element have  …” and in all the cases that are used contractions.

-          PTA and SMA should be defined the first time that appear in the introduction.

-         -  Commercial Optical microscope, SEM, XR diffractometer, VSM, employed should be enumerated.

-         -  In line 136: if Ms is given with an error of “±0.003T” then the value of Ms should be given with one significant number more. As well as in the MS values given in Figure 10.

-         -  I suggest to increase the image quality in Figure 6 and modify the colors selected.

-         - In Figure 11 it is not explained the meaning of “specific magnetization”. If the magnetization is measured in T in Figure 10 why it is not represented in T in Figure 11.

-        -   Images of Figure 7 should be named a-h.

-         -  In all the cases correct “i.e.” by “i.e.,”.

-         -  In line 345:”the magnetic domain wall thickness δw can be estimated to be about 100nm by referring to iron (~0.1μm) [4]”, it is not clear the DW thickness estimation and it also not appears in the refered paper.

-         - In line 348 a reference is missed for the estimated particle size of most “low-Ms”.
